# Discovery and Characterization of Actively Replicating DNA and Retro-Transcribing Viruses in Lower Vertebrate Hosts Based on RNA Sequencing

**DOI:** 10.3390/v13061042

**Published:** 2021-05-31

**Authors:** Xin-Xin Chen, Wei-Chen Wu, Mang Shi

**Affiliations:** School of Medicine, Sun Yat-sen University, Shenzhen 518107, China; xinxinchen9@163.com (X.-X.C.); wuweixiongde@126.com (W.-C.W.)

**Keywords:** metatranscriptomics, virome, codivergence, vertebrate-associated DNA virus

## Abstract

In a previous study, a metatranscriptomics survey of RNA viruses in several important lower vertebrate host groups revealed huge viral diversity, transforming the understanding of the evolution of vertebrate-associated RNA virus groups. However, the diversity of the DNA and retro-transcribing viruses in these host groups was left uncharacterized. Given that RNA sequencing is capable of revealing viruses undergoing active transcription and replication, we collected previously generated datasets associated with lower vertebrate hosts, and searched them for DNA and retro-transcribing viruses. Our results revealed the complete genome, or “core gene sets”, of 18 vertebrate-associated DNA and retro-transcribing viruses in cartilaginous fishes, ray-finned fishes, and amphibians, many of which had high abundance levels, and some of which showed systemic infections in multiple organs, suggesting active transcription or acute infection within the host. Furthermore, these new findings recharacterized the evolutionary history in the families *Hepadnaviridae*, *Papillomaviridae*, and *Alloherpesviridae*, confirming long-term virus–host codivergence relationships for these virus groups. Collectively, our results revealed reliable and sufficient information within metatranscriptomics sequencing to characterize not only RNA viruses, but also DNA and retro-transcribing viruses, and therefore established a key methodology that will help us to understand the composition and evolution of the total “infectome” within a diverse range of vertebrate hosts.

## 1. Introduction

The development of metagenomics and next-generation sequencing technologies has revolutionized the way in which we discover and characterize viruses. These methods provide an unbiased view of the virome within a host, expand our knowledge of viral diversity, and fill in the “gaps” between many higher virus taxa [1,2,3,4]. Among various metagenomic methods used for virus discovery, metatranscriptomics (i.e., total RNA sequencing) represents a simple but efficient approach that not only transforms our view of the genomic diversity of RNA viruses in a wide range of hosts [5,6,7], but also expands our knowledge of the diversity of DNA viruses [8,9,10,11]. Indeed, after reanalyzing the 92 metatranscriptomics data from invertebrate RNA sequencing, Porter et al. [9] were able to reveal DNA viruses from at least 29 families, including 19 novel species from both large and small DNA virus families, and with good genome coverage. Furthermore, with RNA sequencing, some DNA viruses can reach extremely high abundance levels [8], reflecting active RNA transcription and potentially acute infection within the host. On the other hand, metatranscriptomics can also reveal genomic information from viruses that are often known for chronic and latent infections, such as herpesviruses and papillomaviruses [10,11], which highlights the capacity of RNA sequencing for the discovery of viruses other than those with RNA genomes.

Lower vertebrates represent important groups of hosts whose viral diversity was only recently expanded thanks to the metatranscriptomics approach [7,12]. Specifically, RNA viruses were discovered in Agnatha (jawless fishes), Chondrichthyes (cartilaginous fishes), Actinopterygii (ray-finned fishes), Dipnomorpha (lungfishes), Amphibia, and Reptilia through the metatranscriptomics method, leading to the establishment of a long-term virus–host relationship for most of the vertebrate-associated RNA virus groups [7]. Similarly, the knowledge of diversity and host range has also been expanded for the family *Hepadnaviridae*—a group of retro-transcribing (RT) viruses whose divergence from the newly established nackednaviruses (NDV) can be traced back to ~400 million years ago, well before the rise of tetrapods [13]. As for DNA viruses, several viral families/orders have been found to be associated with ray-finned fishes, amphibians, and/or reptiles, including *Parvoviridae* [14,15], *Iridoviridae* [16,17,18], *Herpesvirales* [19,20,21], *Adenoviridae* [22,23,24], *Papillomaviridae* [25,26], *Polyomaviridae* [27,28,29], and *Circoviridae* [30]. Among these, the order *Herpesvirales* shows the widest host range and a relatively straightforward virus–host codivergence relationship [31], whereas for other virus families/orders the understanding of viral diversity in lower vertebrate hosts is still very limited [32]. Furthermore, current discovery of DNA and RT viruses is mainly limited to economically important or ecologically widespread host species, while very few discoveries have been carried out on several key taxonomic groups in the evolution of vertebrates, such as jawless fishes, cartilaginous fishes, and lungfishes, leaving substantial gaps in the understanding of viral evolution.

Given that (1) RNA sequencing is capable of detecting viruses belonging to DNA and RT virus categories, and that (2) there are rich metatranscriptomics and transcriptomics sequencing results available in the public database that cover several key taxa in the lower vertebrates, we carried out a large-scale survey of DNA and RT viruses based on 711 RNA sequencing data that were generated from the previous studies. The results revealed divergent novel viruses, expanding the diversity of existing virus groups while providing key evidence that these are actively transcribing viruses, shedding important new insights on the virus–host relationship for these important virus groups.

## 2. Materials and Methods

### 2.1. Collection of Metatranscriptomics and Transcriptomics Datasets

Our study contained two datasets—(1) the metatranscriptomics dataset (n = 125), which included the libraries used in the study that described the RNA viral diversity in lower vertebrate hosts [7] (Appendix A); and (2) the all assembled transcriptomics dataset (n = 586)—which were downloaded from Transcriptome Shotgun Assembly (TSA) database on 29 November 2020 (ftp://ftp.ddbj.nig.ac.jp/ddbj_database/tsa/, Appendix A). Both datasets were associated with non-mammalian and non-avian vertebrate hosts, and together, they included a total of more than 400 host species from 6 major lower vertebrate classes/superclasses, including superclass Agnatha (jawless fishes), class Chondrichthyes (cartilaginous fishes), class Actinopterygii (ray-finned fishes), class Dipnomorpha (lungfishes), class Amphibia (amphibians), and class Reptilia (reptiles) [7]. Host species information was further confirmed by analyzing the complete or partial cytochrome c oxidase (COI) gene from each sample.

### 2.2. Discovery of DNA and Retro-Transcribing Viruses

For each library, we compared the assembled metatranscriptomics and transcriptomics contigs against the GenBank non-redundant (nr) database using Diamond version 0.9.25.126 [33]. The taxonomic lineage information was collected for the identification of the viruses. To distinguish vertebrate-associated viruses from those infecting parasites, cohabiting microbes, and food, we selected only contigs that were related to known vertebrate-infecting virus families or genera, which included both (1) DNA viruses—namely, parvovirus, circovirus, papillomavirus, and herpesvirus—and (2) RT viruses—namely, hepadnavirus and retrovirus. Subsequently, viral contigs with unassembled overlaps were merged to form longer genome fragments by using the SeqMan program implemented in the Lasergene software package (version 7.0, DNAstar). These merged virus contigs were further confirmed and/or extended by mapping reads to the existing sequences with Bowtie2 [34]. Contigs that had premature stop codons within an expected ORF were considered endogenous virus elements, and were not considered to be virus genomes. Finally, confirmed viral genomic sequences of >500 bp in length were used for further genomic characterization and phylogenetic analyses.

### 2.3. Virus Genome Characterization and Abundance Estimations

For each newly discovered virus genome, open reading frames (ORFs) were predicted using TransDecoder and annotated based on those from the related reference virus genomes. Conserved domains were predicted by comparing the sequence against the conserved domain database (CDD) with an e-value threshold of 1. For large, complex DNA viruses, their genomes were further characterized and visualized using Easyfig v2.2.5 [35]. Other viral genome structures were characterized and compared using Geneious software package 2021.0.3 [36]. To determine the abundance levels of each virus genome/genome segment, we used the statistic “total mapped virus reads per million total reads” (RPM). To estimate the abundance level of herpesvirus transcripts, we used the statistic “reads per kilobase of transcripts, per million mapped reads” (RPKM).

### 2.4. Phylogenetic Analyses

The phylogenetic trees were reconstructed for each vertebrate-associated DNA and RT virus family/order, including *Herpesvirales*, *Parvoviridae*, *Papillomaviridae*, *Circoviridae*, *Hepadnaviridae,* and *Retroviridae*. In addition to those revealed by this study, reference virus genome sequences representative of the diversity in each family were downloaded from the GenBank database. The family-level trees were reconstructed based on one or several conserved protein alignments—namely, the major capsid protein (capsid), ATPase subunit of terminase, and protein kinases for the order *Herpesvirales* (n = 87, 84, and 68, respectively), the non-structural protein (NS1) for the family *Parvoviridae* (n = 51), the viral replicase initiation protein (Rep) for the genus *Circovirus* (n = 29), the late protein (L1) for *Papillomaviridae* (n = 54), the polymerase protein for *Hepadnaviridae* and related viruses (n = 38) [13], and the RNA-dependent DNA polymerase for the family *Retroviridae* (n = 70). Amino acid sequence alignment of these viruses was performed in MAFFT (version 7) [37], and alignment gaps and ambiguously aligned regions were removed using TrimAI(version 1.2) [38] Based on the trimmed alignments, maximum likelihood (ML) phylogenetic trees were re-constructed in PhyML version 3.0 [39], employing the LG amino acid substitution model, and the subtree pruning and regrafting (SPR) branch-swapping algorithm.

### 2.5. Virus–Host Codivergence Analyses

For those confirmed vertebrate-associated viruses identified in this study, we reconciled the phylogeny of each virus family/order and that of their host for the examination of the codivergence hypothesis. The virus phylogenies were based on the family/gene-level maximum likelihood phylogenies estimated in this study, whereas the related host topologies were inferred and downloaded from the TIMETREE website (http://www.timetree.org/, accessed on 15 March 2021). We used the program Jane v4 [40] which is an event-based method that finds the least cost solution to reconcile virus and host phylogenies; specifically, it uses genetic algorithm computing solutions to map a parasite tree onto the host tree with the least cost for five types of possible events, each assigned a specific cost: codivergence = 0, duplication = 1, host switch = 1, loss = 1, failure to diverge = 1. The number of generations and the population size were both set to 100. The significance of codivergence was obtained by comparing the estimated costs to null distributions calculated from 100 randomizations of virus–host tip mapping.

### 2.6. Data Availability 

All virus genome sequences generated in this study have been deposited in GenBank under the accession numbers MZ244208-MZ244223 (pending release) and in Figshare (doi:10.6084/m9.figshare.14569476).

## 3. Results

### 3.1. Discovery and Characterization of DNA and RT Viruses Based on Metatranscriptomics and Transcriptomics Data

For metatranscriptomics sequencing data, our data comprised 6 major lower vertebrate host groups (a total of 125 libraries), including Agnatha (jawless fishes), Chondrichthyes (cartilaginous fishes), Actinopterygii (ray-finned fishes), Dipnomorpha (lungfishes), Amphibia, and Reptilia. From these host groups, four vertebrate-associated DNA and RT virus families were identified—namely, *Circoviridae*, *Herpesvirales*, *Pappillomaviridae*, and *Hepadnaviridae* (Figure 1A). In comparisons, 14 RNA virus families/orders were identified in these libraries from a previous study based on the same dataset [7] (Figure 1A). Among DNA and RT viruses, hepadnaviruses were most frequently detected (14/125 libraries), and appeared in the cartilaginous fishes, ray-finned fishes, and amphibian libraries (Figure 1B). On the other hand, herpesviruses and papillomaviruses appeared only in the cartilaginous fishes and amphibian libraries, respectively (Figure 1B). Despite their low prevalence rate, the abundance level of DNA and RT viruses was very high, with the highest reaching 17,618 and 559 RPM for hepadnaviruses and herpesviruses, respectively (Figure 1C). On the other hand, the abundance levels of papillomaviruses and circoviruses were much lower in the libraries we examined (Figure 1C). Generally, the discovery of DNA/RT viruses via metatranscriptomics sequencing revealed a 14.4% pool positive rate and 13 virus species, which was much lower than that of RNA viruses (Figure 1D,E), and the abundance levels were generally lower for DNA/RT viruses as well, although for some libraries they reached very high levels (Figure 1F).

For transcriptomics (poly-A tailed enrichment) sequencing data, we examined a total of 586 public transcriptomics libraries, which included 352 species of low vertebrate hosts, and detected members of *Hepadnaviridae*, *Parvoviridae*, *Circoviridae*, *Herpesviridae*, *Polymaviridae,* and *Papillomaviridae*. Among these, *Hepadnaviridae* had the highest positive rates (Figure 1G). The overall positive rate for DNA/RT viruses in the transcriptomics libraries was 4.1%.

### 3.2. A Candidate Member of Alloherpesvirus Discovered in Rana Rugulosa

We identified a single species of alloherpesvirus from two metatranscriptomics libraries associated with Chinese tiger frogs (*Hoplobatrachus rugulosus*)—HWWF (lung tissues) and HWWGP (liver and spleen tissues)—to which we temporarily assigned the species name ranid herpesvirus 4 (RHV4). Based on metatranscriptomics sequencing, we were able to recover 6 major contigs, which totaled 85,404 bp in length (non-repeat) and approximately 36.8% of the expected genome length (Figure 2A). The overall abundance levels were 534 and 145 RPM in lung and liver tissues, respectively, suggesting active and systemic infections within the hosts.

We then compared the genome fragments of newly identified RHV4 with its closest relative RHV2, another herpesvirus isolated from northern leopard frogs, and this revealed that they mostly covered the central part of the reference genome (i.e., from ORF68 to ORF131 of RHV2) [20], which included the 12 conserved genes for all herpesviruses [41]. While the two viruses shared similar genome arrangements and encoded genes, the genetic diversity between the encoded proteins varied from 22.6% (ORF50 protein) to 58.6% (ORF68 protein) amino acid identity, a divergence level large enough for assignment of a new species. Furthermore, based on RNA sequencing data, we estimated the expression level (measured in RPKM) for each transcript (Figure 2B), which revealed that abundance was generally higher in structural (e.g., major capsid protein, membrane glycoprotein) and regulatory (e.g., protein kinase) genes than in replication genes (DNA polymerase, helicase), although it is unclear whether this gene expression pattern is associated with a specific stage during the infection.

In order to place the newly identified RHV4 within the context of the order *Herpesvirales*, we reconstructed phylogenetic trees based on three conserved proteins within the order—namely, capsid maturational protease, ATPase subunit of terminase, and protein kinase. The phylogenies showed that the RHV4 was consistently clustered with RHV2, which in turn belonged to the genus *Batrachovirus* of the family *Alloherpesviridae* (Figure 2C).

### 3.3. Identification and Characterization of Small DNA Viruses

We identified five species of virus belonging to the small ssDNA and dsDNA categories from the metatranscriptomics and transcriptomics libraries, including members of the families *Parvoviridae* (n = 2), *Circoviridae* (n = 1), and *Papillomaviridae* (n = 2) (Figure 3, Appendix A). The two potentially novel parvoviruses were identified from *Parargyrops edita* (standing fish) and *Schizothorax prenanti* (Ya-fish), respectively, and we temporarily named them *Parargyrops edita* parvovirus 1 (PaePV1) and *Schizothorax prenanti* parvovirus 1 (ScpPV1), respectively. The two viruses were highly divergent from the other parvovirus strains. For PaePV1, we revealed the partial genome (1221 bp) with the NS1 gene, and it did not belong to any of the existing parvovirus genera (Figure 3A), while the closest parvovirus strain (*Dependoparvovirus* sp.) shared only 35.4% amino acid identity. The second virus, ScpPV1, which had a 3741-bp-long contig covering both the NS1 and VP proteins, was clustered with the *Syngnathus scovelli* chapparvovirus (51.9%), and belonged to the genus *Ichthamaparvovirus*, subfamily *Hamaparvovirinae* (Figure 3A).

One circovirus-related contig (1757 bp) was identified from a transcriptomics sequencing of liver and brain tissues collected from European eel (*Anguilla anguilla*). This virus contained the predicted ORF, replicase, and shared 95.8% and 96.2% of its amino acid identity with *Anguilla anguilla* circovirus and *Anguilla anguilla* circovirus 2 [42,43], which were also identified from *Anguilla anguilla*. Therefore, we temporarily named this virus *Anguilla anguilla* circovirus 3 (AACV 3). Furthermore, phylogenetic analyses revealed that fish-related viruses, including *Anguilla anguilla* circovirus [42], *Anguilla anguilla* circovirus 2 [43], Barbel circovirus [44], *Anguilla anguilla* circovirus 3, and *Silurus glanis* circovirus isolate H5 [45], formed clusters that were relatively divergent to the classic mammalian and avian circoviruses (Figure 3B). 

Two novel species of papillomaviruses were detected in a single metatranscriptomics library from the liver and gills tissues of *Sparus aurata*, a cartilaginous fish, which we named *Urolophus aurantiaus* papillomavirus 1 (UaurPV 1) and *Urolophus aurantiaus* papillomavirus 2 (UaurPV 2), respectively. Among these, the contigs associated with UaurPV 1 had only L1 genes, whereas those of UaurPV 2 had E1 and L1 genes. The two viruses were closely related to one another, but were extremely divergent (<47.3% amino acid identity) from the existing member of the family *Papillomaviridae*. Phylogenetic analyses revealed that UaurPV 1 and 2 formed a highly divergent clade in the papillomavirus tree, basal to all of the papillomaviruses identified from Tetrapoda as well as ray-finned fishes (Figure 3C).

### 3.4. Identification of RT Viruses and Endogenous Virus Element of RT Viruses

In cartilaginous fishes, ray-finned fishes and amphibians, we identified, based on metatranscriptomics data and transcriptomics data, a total of five complete genomes and seven partial genomes of hepadnaviruses that were highly divergent from the existing members of the hepadnavirus family previously detected in other vertebrate hosts (Appendix A). Our study represents the first time that viruses were identified from cartilaginous fishes—namely, ringstreaked guitarfish (*Rhinobatos hynnicephalus*) and rat fish (*Chimaera phantasma*)—which were all recovered from the liver samples, suggesting that the associations between HBV and vertebrates are now extended to cartilaginous fishes. Among them, the complete genome (3435 bp) of hepadnavirus was identified from ringstreaked guitarfish, which was named as *Rhinobatos hynnicephalus* hepatitis B virus (i.e., RHHBV). The average coverage of RHHBV was 802 folds (407.5 RPM), which encoded the polymerase, core, and surface proteins, and shared an amino acid similarity of 37.2% with the roundleaf bat hepatitis B virus (Appendix A). On the phylogeny, the newly identified HBVs were clustered into three distinctive lineages: viruses identified from cartilaginous fishes (i.e., RHHBV and CPHBV) formed a monophyletic cluster with majority of the viruses identified from ray-finned fishes (i.e., *Anguilla anguilla*, *Astatotilapia sp*, *Astyanax mexicanus*, and *Chionodraco hamatus*), and together they formed a fish-associated cluster (i.e., metahepadnaviruses), sister to the mammalian HBVs; Anguilliformes HBV 3 was clustered with the white sucker hepadnavirus to form a second fish-associated HBV cluster (i.e., parahepadnaviruses) enveloped within the hepadnaviruses clade; the three viruses identified from amphibians (i.e., *Odorrana tormota*, *Bufo gargarizans*, and *Hoplobatrachus rugulosus*) formed a cluster with another amphibian virus, Tibetan frog HBV, and together they formed a sister clade (i.e., herpetohepadnavirus) to HBV circulating in birds (Figure 4). No nackednavirus [13] was identified from our metatranscriptomics sequencing or transcriptomics data. Interestingly, despite their positions within the family *Hepadnaviridae* and the presence of PreS/S domains, the *Rhinobatos hynnicephalus* HBV, *Chimaera phantasma* HBV, and *Hoplobatrachus rugulosus* HBV viruses showed signs of systemic infection, rather than a marked liver tropism as observed in hepadnaviruses infecting tetrapods [46]. Indeed, these viruses exhibited relatively high viral abundance levels in multiple organs other than the liver, such as the gut and the gills.

Our BLAST search also revealed a number of contigs carrying reverse transcriptase (RT), which were related to those from the family *Retroviridae*. These contigs were identified from a diverse range of hosts, including cartilaginous fishes (n = 2), ray-finned fishes (n = 2), amphibians (n = 46), and reptiles (n = 25), and some of them had been described in previous studies [47,48]. Despite their extensive diversity, the majority of the viruses discovered in this study had incomplete genomes or disrupted ORFs, so these were most likely expressed endogenous viruses [48]. Nevertheless, we did observe some contigs that contained the complete “gag–pol–env” gene set (Figure 5), and these might present as exogenous viruses, although it is unlikely to confirm based on current data whether these elements had viable viral particles. 

### 3.5. Testing the Codivergence Relationship between the DNA Viruses and Their Vertebrate Hosts

We first compared the phylogenies of viruses and hosts for *Herpesvirales* (vertebrate-associated clade), *Papillomaviridae*, *Hepadnaviridae,* and *Parvoviridae* (vertebrate-associated clade). With the exception of *Parvoviridae*, the rest of the virus families/orders showed striking similarity between virus and host trees (Figure 6A, Appendix A), which was reflected in the observation that (1) there was obvious clustering of viruses based on host taxonomy—namely, Chondrichthyes, Actinopterygii, Amphibia, Aves, Reptilia, and Mammalia—and (2) that the relationships between major groups of viruses reflected those of their hosts (Figure 6A). Therefore, to further examine the relationship between vertebrate DNA/RT viruses and their hosts, we performed codivergence tests for the four virus families/orders. While the results suggested a codivergence relationship for all four families/orders (*p* < 0.01), there were in some cases relatively more frequent host switch events for parvoviruses and papillomaviruses (Figure 6B), suggesting occurrence of occasional host switch events in the background of virus–host codivergence.

## 4. Discussion

In this study, we discovered a total of 18 lower-vertebrate-associated DNA and RT viruses from 711 RNA sequencing results downloaded from the SRA and TSA databases, among which most discoveries of more complete virus genomes were based on metatranscriptomics approaches. While some of the RT viruses (namely, hepadnaviruses) can have complete genome coverage through RNA sequencing, DNA viruses often have disjointed or partial genome coverage, even with high viral abundance levels. This is expected given the presence of non-coding (and hence non-transcribing) regions, as well as the huge variation in the expression levels of different genes in the DNA virus genomes. Despite that, metatranscriptomics is able to reveal the core set of functionally conserved viral genes in DNA viruses (Figure 2A), which is crucial for the following evolutionary and biological characterizations. Collectively, these findings confirm that metatranscriptomics is a useful method for the discovery and characterization of viruses other than those with RNA genomes.

With metatranscriptomics sequencing, the pool discovery rate for vertebrate-associated DNA viruses (10.4%) is much lower in comparison to that for RNA viruses (84.2%). This is expected, because with RNA sequencing, it can only reveal DNA viruses that undergo active transcription, and therefore is not an exhaustive collection of all DNA viruses present within the sample. Indeed, a study that compares metatranscriptomics and viral particle enrichment approaches [49,50,51,52] on the same set of samples revealed that the metatranscriptomics approach discovered less DNA viruses in general, and for those revealed by both approaches, the metatranscriptomics approach tended to reveal lower abundance levels [53]. Nevertheless, although it might underestimate the diversity and prevalence rate of DNA viruses within the hosts, metatranscriptomics tends to reveal DNA viruses that are going through active biosynthesis and replications instead of those with latent infections, and this could be strong indication that the viruses discovered are likely to be associated with active or even acute infections within the hosts.

To avoid false positives in virus discovery and incorrect inferences on virus–host associations, we limited the scope of our search to vertebrate-associated viruses, which excluded the majority of the viruses that are associated with reagent contamination [54,55], undigested food, parasites, or co-inhabiting organisms. Furthermore, several newly discovered viruses showed systemic infections (Figure 2B) and/or phylogenetic clustering to viruses from related host groups (Figure 3 and Figure 4), and therefore further confirmed that these viruses were associated with their principal hosts. In addition to host associations, we also considered the possibility that the newly identified sequences were EVEs that were incorporated into the host genomes [56]. While it is unlikely to definitely exclude the possibility of EVEs based on current data alone, the sequences we described here are all characterized with intact and undisrupted coding regions of key functional domains, which is often not the case for most EVEs. 

Our study revealed a novel alloherpesvirus (RHV4) in Chinese tiger frogs. The virus had relatively high abundance levels (up to 559 and 224 RPM) in pooled (24 individuals) lung and liver tissue samples, suggesting active infection that potentially involves multiple individuals within the population. Before this study, four herpesviruses had been identified in amphibians—namely, ranid herpesvirus 1, 2, and 3 (RHV1, 2, and 3) and bufonid herpesvirus 1 [20,41,57]. Among these, RHV1 is associated with renal adenocarcinomas [58,59,60,61,62], RHV3 is known for causing skin infections [41], whereas RHV2 is unclear of any disease association, although it is isolated from the urine of tumor-bearing *Rana pipiens* [63], and bufonid herpesvirus 1 is associated with severe dermatitis in *Bufo bufo* [57]. Unfortunately, no clinical data were available for the Chinese tiger frog population that harbors RHV4. Nevertheless, our metatranscriptomics results suggest that it may cause systemic infections that are characterized by active replications in multiple tissues or organs. 

Furthermore, it is striking that metatranscriptomics alone can reveal more than one-third of the expected genome length for large DNA viruses. Indeed, the RHV4 genome structures revealed by metatranscriptomics largely followed those of its closest relative, RHV2 (Figure 2A). However, such resemblance was only observed at the central part of the RHV2 genome, whereas no assembled transcripts were found to share sequence homology to genes at the 5′ end (~62,457 bp) or 3′ end (~36,087 bp) of the RHV2 genome (Figure 2A). One possible explanation for this is that the majority of the conserved and colinear genes shared by RHV1/4 are located at the central part of their genomes, whereas few homologies can be found towards the genomes’ ends [41]. Regardless, further isolation and sequencing of the RHV4 genomes is required for subsequent comparisons between different ranid herpesviruses.

Our study also revealed a total of 12 divergent hepadnaviruses from Chondrichthyes, Ostrichthyes, and Amphibians—among which 5 had complete genomes—and therefore greatly expanded the diversity and host range of this family. In a previous study, Lauber et al. [13] discovered a highly divergent nackednavirus group related to hepadnaviruses but lacking the envelop proteins, and this was found exclusively in ray-finned fishes. The discovery of this new family set the evolutionary timescale of hepadna-like viruses to more than 400 million years ago, concurrent to the divergence between actinopterygians and sarcopterygians [13]. However, our discoveries suggest an even more ancient origin of hepadna-like viruses. This is because the diversity expansion in this study, mostly by viruses found in ray-finned fishes and cartilaginous fishes, was mainly associated with enveloped hepadnaviruses in the “metahepadnaviruses” and “parahepadnaviruses” groups. As a result of this diversity expansion, these two groups were unlikely to have simply resulted from “secondary invasions” of actinopterygians through host switches from sarcopterygians. Indeed, based on our estimations, the codivergence points were now placed within the enveloped hepadnaviruses clade rather than between enveloped hepadnavirus and nackednaviruses.

Recent pathogen discovery works have also witnessed an expansion of the host range for small DNA viruses—such as papillomaviruses, polyomaviruses, circoviruses, and parvoviruses—which was previously dominated by mammalian viruses, but now includes increasing numbers of lower vertebrate hosts [14,25,26,42,43,44,45,64,65]. Our findings further expanded this range to ray-finned fishes for *Parvoviridae*, and to cartilaginous fishes for *Papillomaviridae*; the latter is represented as the most divergent cluster in the tree (Figure 3), an observation that bears striking similarity to the evolutionary history of vertebrate hosts. Given that viruses discovered in ray-finned fishes and cartilaginous fishes all formed basal or sister lineages to those discovered in tetrapods, this is highly compatible with the hypothesis of virus–host codivergence, which in turn sets a long-term evolutionary timescale for these virus families (Figure 6). Nevertheless, our discoveries so far by no means represent the full diversity of DNA viruses in lower vertebrates, since the host species we investigated are just the tip of an iceberg. With that, our results underline the importance of virus discoveries in a diverse range of vertebrate hosts in revealing the whole picture of vertebrate virus evolution.

## Figures and Tables

**Figure 1 viruses-13-01042-f001:**
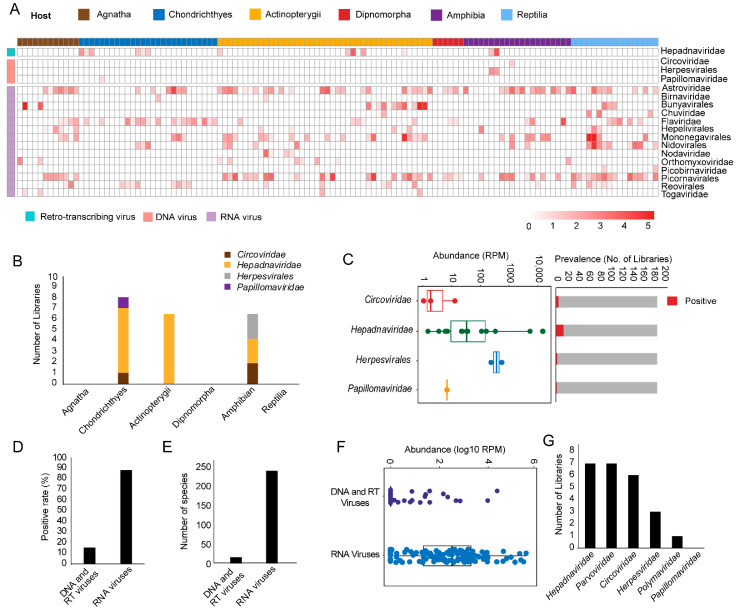
Discovery and characterization of vertebrate-associated DNA, RT, and RNA viruses in lower vertebrate hosts. (**A**) Heat map showing the presence and abundance of major DNA, RT, and RNA virus groups discovered from the 125 metatranscriptomics sequencing data belonging to 6 major lower vertebrate host groups. The RNA virus genomes used here were derived from the 2018 study [7]. (**B**) Bar plot showing the types and positive rates of DNA/RT viruses discovered in each host group. (**C**) Left: Box plot and scatter plots showing the abundance distribution of DNA/RT virus families discovered in this study; each box has the upper, median, and lower quartiles, and each circle point represents one library; Right: bar plot showing the positive rate for each of the virus families. (**D**) Comparisons of the pool positive rate between DNA/RT and RNA viruses. (**E**) Comparisons of the number of DNA/RT and RNA virus species detected. (**F**) Box plot and scatter plots showing the distributions of abundance levels of DNA/RT (purple) and RNA viruses (blue), respectively. (**G**) The distribution of the number of transcriptomics libraries positive for DNA/RT viruses.

**Figure 2 viruses-13-01042-f002:**
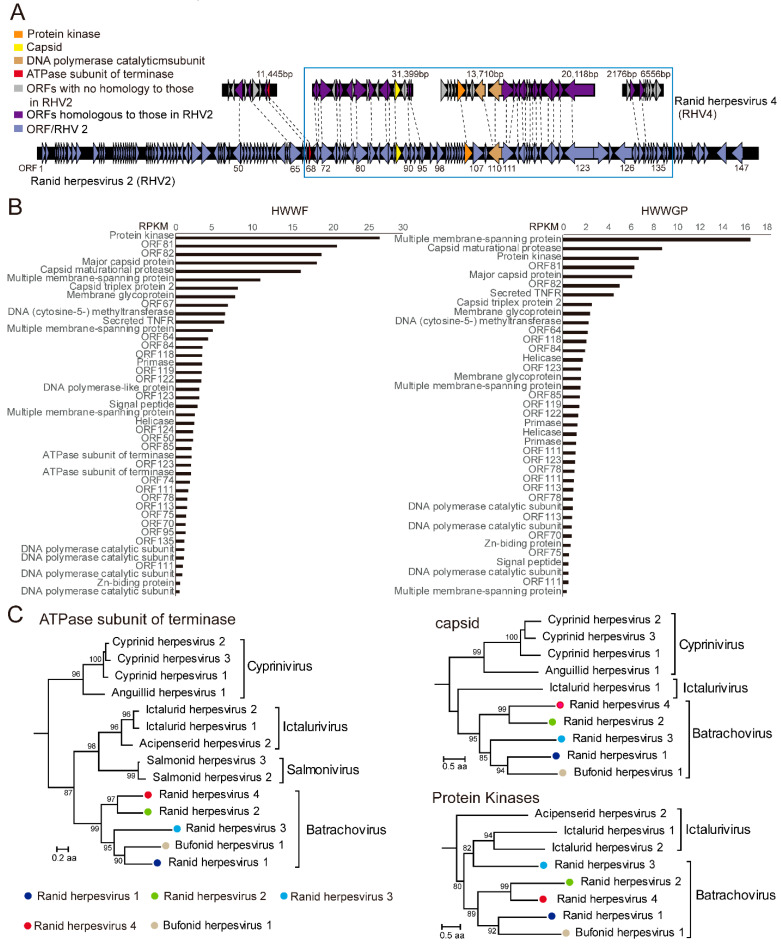
Genomic, transcriptomics, and phylogenetic analyses of the newly identified ranid herpesvirus 4. (**A**) Comparison of the genome structures of ranid herpesvirus 2 (RHV2, NC_008210.1) and RHV4. Genes are shown as rectangles with arrow tips showing the transcription direction, and those conserved in both viruses are connected with dotted lines. (**B**) Bar graph showing the expression levels of different genes by RHV4 in lung (HWWF) and liver/spleen (HWWGP) tissues. The horizontal axis represents the abundance of each gene as measured by RPKM. (**C**) The maximum likelihood phylogenetic trees based on the capsid maturational protease (capsid), ATPase subunit of terminase, and protein kinase proteins, respectively, showing the positions of RHV4 (solid red circle) within the diversity of the family *Alloherpesviridae*. Four amphibian-associated herpesviruses discovered previously are marked in different colors. All trees are midpoint rooted, and the taxonomy information is labelled to the right of each tree.

**Figure 3 viruses-13-01042-f003:**
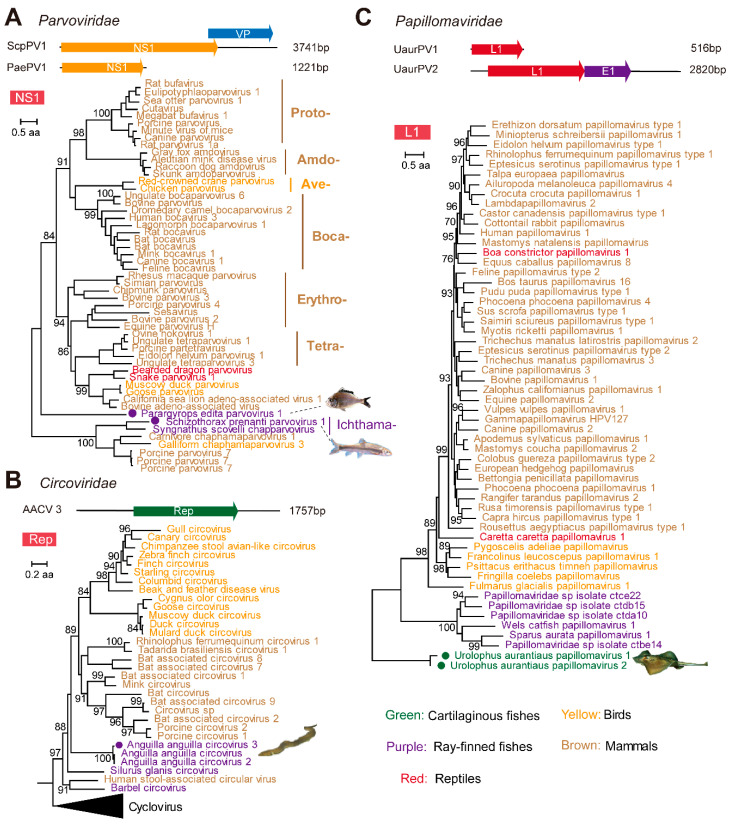
Genomic structures and evolutionary histories of novel small DNA viruses detected from metatranscriptomics and transcriptomics libraries. These include five potential new viruses from three families—namely (**A**) *Parvoviridae*, (**B**) *Circoviridae*, and (**C**) *Papillomaviridae*. The phylogenetic trees were reconstructed based on non-structural protein 1 (NS1), replicase protein, and L1 protein, respectively. All trees are midpoint rooted and scaled to the number amino acid substitutions per site. The virus names are color coded to reflect their host group. The positions of newly discovered viruses are shown in solid red circles in each tree and the host. Branch support values (>70%) are shown at the key nodes. All of the genomic structures are shown above the corresponding phylogenetic tree and the potential proteins or protein domains they encode are labeled in the predicted ORFs of these genomes.

**Figure 4 viruses-13-01042-f004:**
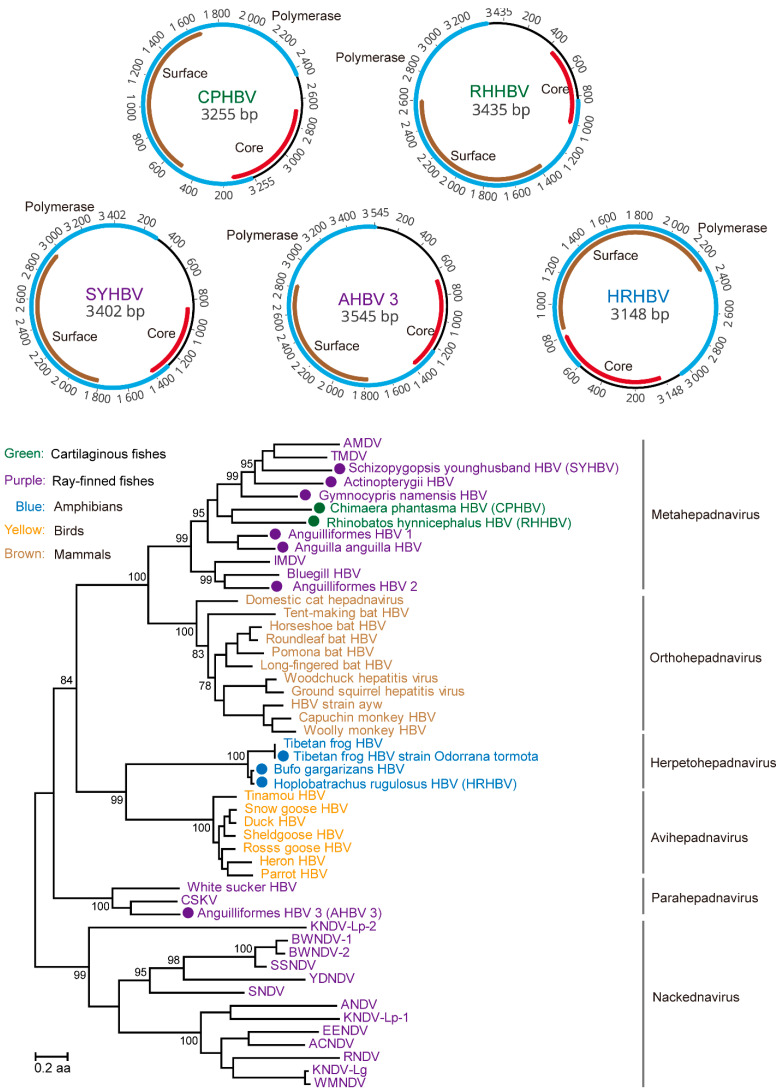
Genome organization and evolutionary analyses of hepadnaviruses discovered from metatranscriptomics and transcriptomics data. Upper panel: Genome structures of five complete hepadnavirus genomes, including *Rhinobatos hynnicephalus* HBV (RHHBV), *Chimaera phantasma* HBV (CPHBV), *Anguilliformes* HBV 3 (AHBV 3), *Schizopygopsis younghusbandi* HBV (SYHBV), and *Hoplobatrachus rugulosus* HBV (HRHBV); the predicted coding sequences are shown in blue (polymerase), brown (surface), and red (core). Lower panel: Maximum likelihood phylogenetic tree including all 12 hepadnaviruses within the context of enveloped and non-enveloped hepadna-like viruses. Tree is midpoint rooted and scaled to the number of amino acid substitutions per site. Viruses discovered in our study are shown in solid circles in each tree. Branch support values (>70%) are shown at the key nodes. The virus names and dots are color coded to reflect their host group.

**Figure 5 viruses-13-01042-f005:**
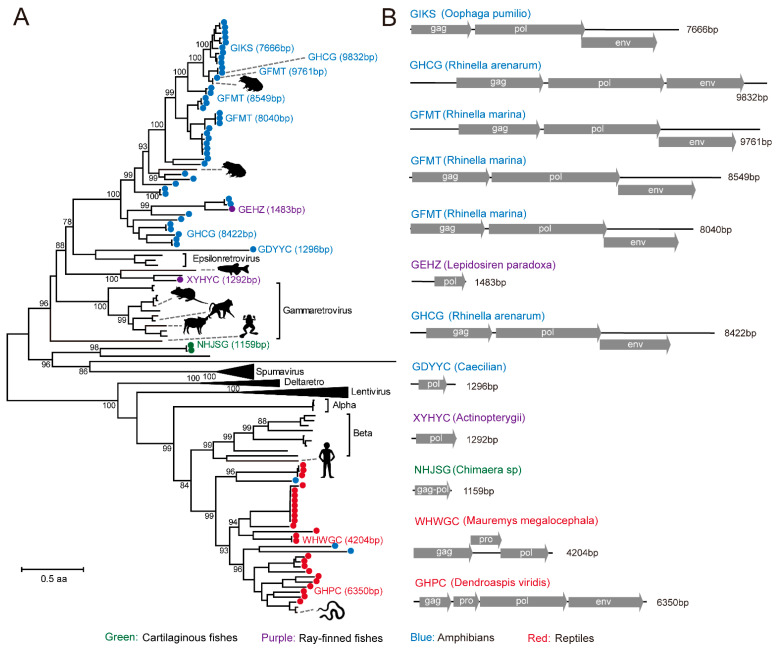
Genome structure and evolutionary analysis of retrovirus-like elements discovered from metatranscriptomics and transcriptomics data. (**A**) Maximum likelihood phylogenetic tree based on newly discovered retro-like virus elements and members of the family *Retroviridae*. (**B**) The structures of ORFs predicted from the retro-like virus elements in lower vertebrate hosts. Tree is midpoint rooted and scaled to the number of amino acid substitutions per site. Newly discovered EVEs are shown in sold circles in each tree, which are color coded to reflect the host group of each virus or EVE. Branch support values (>70%) are shown at the key nodes.

**Figure 6 viruses-13-01042-f006:**
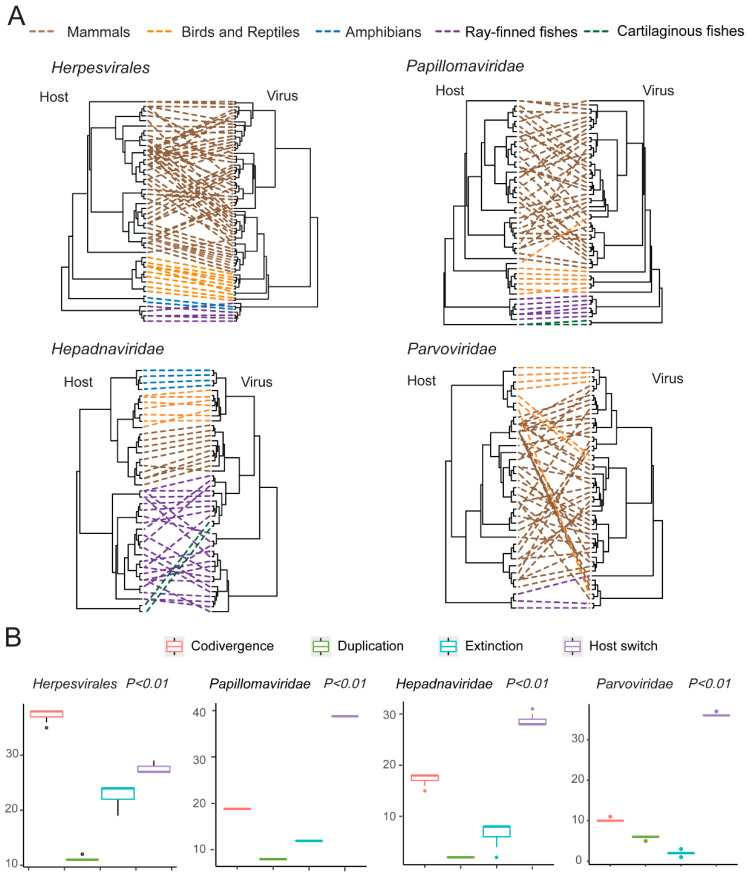
Codivergence relationships between vertebrate hosts and their associated DNA/RT viruses. (**A**) Comparison of the virus and host phylogenetic structures for herpesviruses, papillomaviruses, hepadnaviruses, and parvoviruses. More detailed presentations containing taxon labels are shown in Appendix A. (**B**) The box plot showing the estimation of codivergence events across the phylogeny of vertebrate-associated DNA viruses. Each box plot has the estimated median (center line) and upper and lower quartiles (box limits).

## Data Availability

All virus genome sequences generated in this study have been deposited in GenBank under the accession numbers MZ244208-MZ244223 and in Figshare (doi:10.6084/m9.figshare.14569476).

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
