# Peer review of "Discovery and Characterization of Actively Replicating DNA and Retro-Transcribing Viruses in Lower Vertebrate Hosts Based on RNA Sequencing"

_viruses, 2021, doi:10.3390/v13061042_

Round 1
Reviewer 1 Report
In the manuscript entitled “Discovery and characterization of actively replicating DNA and retro-transcribing virus in lower vertebrate hosts based on RNA sequencing” the authors provide the complete genome or the “core gene sets” of 18 vertebrate-associated DNA and retro-transcribing viruses in cartilaginous fish, ray-finned fish, and amphibians. Generally, the paper is well presented and argued, the conclusions drawn by the authors based on their findings are logical.
Editing of English language and style is required!
Comments:
line 3: correct virus to viruses
line 18: Please reword this sentence “Furthermore, these new findings re-shaped….” since the long-term virus-host co-divergence of the members of these virus families were already published.
line 27: correct has to have
line 28: correct virus to viruses
line 30: correct taxonomies to taxons
line 146: You have to provide the sequences of the novel viruses for the reviewers as a supplementary file if they are not public in the GenBank yet.
line 185: please reword the title of the paragraph
line 197: please cite the following paper: PMID: 17098965, this one describes the genome of RaHV-2
Figure 2C: These phylogenetic trees are too small in size and not informative enough! All herpesviruses of lower vertebrates belong to the family Alloherpesviridae, so you should remove from the tree the viruses of higher vertebrates (Alpha-, Beta-, Gammaherpesviruses) and include members of all genera (Batrachovirus, Cyprinivirus, Salmonivirus, Ictalurivirus) of the family Alloherpesviridae.
Correct the Betrechovirus to Batrachovirus in the figure!
line 241: Please add references regarding barbel circovirus: PMID: 21525210 sheatfish circovirus: PMID: 22426897 and eel circoviruses: PMID: 24991738 and PMID: 28605966 . In the latter article the Anguillid circovirus 2 have already been described (GenBank Acc. No.: KU951580), please rename your virus to Anguillid CV-3.
Figure 3C: The tree contains only the Sparus aurata papillomavirus. There are few more fish papillomaviruses published (PMID: 32014111), these ones should be included in the tree as well. GenBank Acc Nos: MN515404, MH617579, MH617143, MH616908 and MH510267.
Figure 6A: These trees are not informative, please add at least the major taxons in the trees! And enlarge the size of the trees!
Line 384: Bufonid herpesvirus 1 was also described from amphibians (PMID: 30283010).
Discussion: There is only one sentence about the newly discovered parvoviruses, circovirus and papillomaviruses. Please discuss them in more details.
References: number 22 and 24 references are the same!
Supplementary table 1: Batrachovirus and Circovirus should be written with capitals! And correct Betrechovirus to Batrachovirus!
Author Response
Reviewer #1 (Comments and Suggestions for Authors):
In the manuscript entitled “Discovery and characterization of actively replicating DNA and retro-transcribing virus in lower vertebrate hosts based on RNA sequencing” the authors provide the complete genome or the “core gene sets” of 18 vertebrate-associated DNA and retro-transcribing viruses in cartilaginous fish, ray-finned fish, and amphibians. Generally, the paper is well presented and argued, the conclusions drawn by the authors based on their findings are logical.
Editing of English language and style is required!
Comments:
line 3: correct virus to viruses
Response: Changed as suggested.
line 18: Please reword this sentence “Furthermore, these new findings re-shaped….” since the long-term virus-host co-divergence of the members of these virus families were already published.
Response: We agree with the reviewer and replace the “established” with “confirming”; and “re-shaped” with “re-characterized”; in lines 18-19
line 27: correct has to have
Response: Changed as suggested.
line 28: correct virus to viruses
Response: Changed as suggested.
line 30: correct taxonomies to taxons
Response: Changed as suggested.
line 146: You have to provide the sequences of the novel viruses for the reviewers as a supplementary file if they are not public in the GenBank yet.
Response: All sequences generated in this study are now shared through Figshare through the DOI: 10.6084/m9.figshare.14569476, which is also described in “2.6 data availability”.
line 185: please reword the title of the paragraph
Response: We replace the title “A candidate member of herpesvirus in Rana rugulosa” with “A candidate member of alloherpesvirus discovered in Rana rugulosa”, which was more accurate.
line 197: please cite the following paper: PMID: 17098965, this one describes the genome of RaHV-2
Response: Added as suggested.
Figure 2C: These phylogenetic trees are too small in size and not informative enough! All herpesviruses of lower vertebrates belong to the family Alloherpesviridae, so you should remove from the tree the viruses of higher vertebrates (Alpha-, Beta-, Gammaherpesviruses) and include members of all genera (Batrachovirus, Cyprinivirus, Salmonivirus, Ictalurivirus) of the family Alloherpesviridae.
Correct the Betrechovirus to Batrachovirus in the figure!
Response: We replace the original phylogenetic tree with one showing only Alloherpesviridae as suggested. Detailed taxon names are labelled to the right of the tree, which included members of all genera. And typos are also corrected.
line 241: Please add references regarding barbel circovirus: PMID: 21525210 sheatfish circovirus: PMID: 22426897 and eel circoviruses: PMID: 24991738 and PMID: 28605966. In the latter article the Anguillid circovirus 2 have already been described (GenBank Acc. No.: KU951580), please rename your virus to Anguillid CV-3.
Response: The name of newly discovered circovirus is changed to Anguilla anguilla circovirus 3 as suggested. References are added as well.
Figure 3C: The tree contains only the Sparus aurata papillomavirus. There are few more fish papillomaviruses published (PMID: 32014111), these ones should be included in the tree as well. GenBank Acc Nos: MN515404, MH617579, MH617143, MH616908 and MH510267.
Response: We thank the reviewer for the suggestion, 5 divergent fish-associated papillomaviruses have been added in the tree.
Figure 6A: These trees are not informative, please add at least the major taxons in the trees! And enlarge the size of the trees!
Response: We add 4 supplemental figures (Fig S1-4) that contain detailed virus and host information in the trees.
Line 384: Bufonid herpesvirus 1 was also described from amphibians (PMID: 30283010).
Response: Added as suggested.
Discussion: There is only one sentence about the newly discovered parvoviruses, circovirus and papillomaviruses. Please discuss them in more details.
Response: We have re-written the whole paragraph in the discussion section, to read:
“Recent pathogen discovery works have also witnessed an expansion of host range for small DNA viruses such as papillomaviruses, polyomaviruses, circoviruses and parvoviruses which is previously dominated by mammalian viruses but now includes increasing amount of lower vertebrate hosts [14,25-26,42-45,64-65]. Our findings further expanded this range to ray-finned fish for Parvovirinae, and to cartilaginous fish for Papillomaviridae, the latter is represented as the most divergent cluster in the tree (Figure 3), an observation that bears striking similarity to the evolutionary history of vertebrate hosts. Given that viruses discovered in ray-finned fish and cartilaginous fish all formed basal or sister lineages to those discovered in tetrapods, it is highly compatible with the hypothesis of virus-host co-divergence, which in turn sets a long-term evolutionary timescale for these virus families (Figure 6). Nevertheless, our discoveries so far by no means represent the full diversity for DNA viruses in lower vertebrates, since the host species we investigated are just a tip of an iceberg. With that, our results underline the importance of virus discoveries in diverse range of vertebrate hosts in revealing the whole picture of vertebrate virus evolution.”
References: number 22 and 24 references are the same!
Response: We apologize for the mistake here and it has been corrected.
Supplementary table 1: Batrachovirus and Circovirus should be written with capitals! And correct Betrechovirus to Batrachovirus!
Response: Corrected as suggested.

Reviewer 2 Report
The manuscript by Chen et al concerns an analysis of virus sequences belonging to either of six families of DNA and retro-transcribing viruses in transcriptome datasets of lower vertebrates published by others, which include primitive fish species, amphibians and reptiles. Overall, the authors have done an impressive amount of work to retrieve and assemble the data and have performed and extensive, state-of the-art analysis of the viral sequences mined from such databases.
Line 83-84: ‘Host species information was further confirmed by the analyzing the complete or partial cytochrome c oxidase (COI) gene from each sample.’ Can nuclear integrations of mtDNA confound host analysis in these lower vertebrates? Please explain
Line 128: ‘employing the general time-reversible (GTR) substitution model’. Please explain why this model was chosen
Line 167: ‘lower than that of RNA viruses (Figure 1D and E).’ The RNA virus data derive from an earlier, already published investigation, probably by the same authors, but it is not clear from the figures that the authors are re-using data, and did not generate them de novo for the current study. Which is somewhat confusing. Please comment.
Lines 302-305: ‘none of the viruses discovered in this study fell within the exogenous virus clusters, namely, epsilonretrovirus, gammaretrovirus, betaretrovirus, and spumavirus, suggesting that these sequences were most likely endogenous virus elements (EVEs) (Figure 5).’ I do not understand this conclusion. There are endogenous, complete proviruses of each of these retroviral classes in many animal species, and clustering alone cannot readily distinguish between an endogenous or exogenous state unless more data are provided on other individuals of a species, or on tissue distribution. It is likely that the upper blue dots in Fig. 5 represent epsilon-like retroviruses in amphibians, which is a new discovery. Other important aspects to distinguish EVE’s from infectious variants is whether or not your contigs contain complete genomes (LTR-gag-pol-env-LTR), and whether or not the reading frames in these complete genomes are open. If they are not, the provirus is likely an EVE, but if they are open, one cannot be sure. From Fig. 5, I gather that some of your genomes are indeed EVE’s as they are merely fragmented genomes, but GIKS and GFMT may contain complete proviral genomes. Please give more details on your assembled RT-containing contigs. And, at present there is no infectious betaretrovirus in humans, despite the cartoon in Fig. 5, so there may be more endogenous retrovirus sequences in your tree.
Although the English language is generally adequate, the manuscript is full of typos and grammatical errors, such as:
Line 51: can be trace back
Line 121: the non-structure protein
Line 127: maximum-likelinood
Line 177: the mata-transcriptom
Line 205-206: with specific phase
Papillomavirus is sometimes spelled as papillomavirus.
Please revise carefully. And, check also the figures and their legends (Reptilias, for instance in Fig. 6)
Author Response
Reviewer2:
Comments and Suggestions for Authors
The manuscript by Chen et al concerns an analysis of virus sequences belonging to either of six families of DNA and retro-transcribing viruses in transcriptome datasets of lower vertebrates published by others, which include primitive fish species, amphibians and reptiles. Overall, the authors have done an impressive amount of work to retrieve and assemble the data and have performed and extensive, state-of the-art analysis of the viral sequences mined from such databases.
Line 83-84: ‘Host species information was further confirmed by the analyzing the complete or partial cytochrome c oxidase (COI) gene from each sample.’ Can nuclear integrations of mtDNA confound host analysis in these lower vertebrates? Please explain
Response: mtDNA genes integrated into the nuclear genome are called nuclear mitochondrial DNA segments (numts). The COI genes identified can be distinguished from numt by the following features: (i) they were identified from RNA sequencing and were highly expressed; (ii) they contain uninterrupted open reading frame based on mitochondrial genetic code, and many genes were franked by other mitochondrial genes; (iii) for the meta-transcriptomics data set, the host identification is based on both mitochondrial gene sequence as well as morphological identification. In other words, mitochondrial gene sequence is just a confirmation of host species.
Line 128: ‘employing the general time-reversible (GTR) substitution model’. Please explain why this model was chosen
Response: We apologize for the mistake here and we have corrected the base substitution model (GTR) to the amino acid substitution model (LG).
Line 167: ‘lower than that of RNA viruses (Figure 1D and E).’ The RNA virus data derive from an earlier, already published investigation, probably by the same authors, but it is not clear from the figures that the authors are re-using data, and did not generate them de novo for the current study. Which is somewhat confusing. Please comment.
Response: Although the RNA viruses were discovered from 2016 study, the quantification which generated the heat map is carried out in this study. We revised the legend of figure 1 accordingly to reflect this point: “…The RNA virus genomes used here were derived from the 2018 study [7]”.
Lines 302-305: ‘none of the viruses discovered in this study fell within the exogenous virus clusters, namely, epsilonretrovirus, gammaretrovirus, betaretrovirus, and spumavirus, suggesting that these sequences were most likely endogenous virus elements (EVEs) (Figure 5).’ I do not understand this conclusion. There are endogenous, complete proviruses of each of these retroviral classes in many animal species, and clustering alone cannot readily distinguish between an endogenous or exogenous state unless more data are provided on other individuals of a species, or on tissue distribution. It is likely that the upper blue dots in Fig. 5 represent epsilon-like retroviruses in amphibians, which is a new discovery. Other important aspects to distinguish EVE’s from infectious variants is whether or not your contigs contain complete genomes (LTR-gag-pol-env-LTR), and whether or not the reading frames in these complete genomes are open. If they are not, the provirus is likely an EVE, but if they are open, one cannot be sure. From Fig. 5, I gather that some of your genomes are indeed EVE’s as they are merely fragmented genomes, but GIKS and GFMT may contain complete proviral genomes. Please give more details on your assembled RT-containing contigs. And, at present there is no infectious betaretrovirus in humans, despite the cartoon in Fig. 5, so there may be more endogenous retrovirus sequences in your tree.
Response: We agree with the reviewer that there was some haste in the conclusion: those presented with complete “gag-pol-env” do have the potential to be exogenous virus, although we cannot confirm this based on current data (RNA sequencing) and it required DNA/amplicon sequencing and virus isolation to confirm the presence of LTR and viral particles, respectively. Nevertheless, we have made the corresponding revision based on reviewer’s suggestion, to read
“Our blast search also revealed a number of contigs carrying reverse transcriptase (RT) that were related to those from the family Retroviridae. These contigs were identified from a diverse range of hosts, including cartilaginous fish (n = 2), ray-finned fish (n = 2), amphibians (n = 46) and reptiles (n = 25), and some of them had been described in previous studies [47,48]. Despite of their extensive diversity, none majority of the viruses discovered in this study have incomplete genome or disrupted ORFs so that these are most likely expressed endogenous viruses [48]. Nevertheless, we did observe some contigs which contained the complete “gag-pol-env” gene set (Figure 5), so that these may present as exogenous form, although it is unlikely to confirm based on current data whether these elements had viable viral particles.”
We have also added more “complete” genomes to the figure as requested by the reviewer.
Although the English language is generally adequate, the manuscript is full of typos and grammatical errors, such as:
Line 51: can be trace back
Line 121: the non-structure protein
Line 127: maximum-likelinood
Line 177: the mata-transcriptom
Line 205-206: with specific phase
Papillomavirus is sometimes spelled as papillomavirus.
Please revise carefully. And, check also the figures and their legends (Reptilias, for instance in Fig. 6)
Response: We apologize for this and we have gone through the manuscript a few more times to correct any typos and grammatical errors left in the context.

Round 2
Reviewer 1 Report
The paper is suitable for publication.
Please check the format of the newly added references!
Ref 42: the volume and the page numbers are missing, the first authors given name should be replaced with its initial.
Ref 43-44-45 and 65: All initials should be after the surnames.
Ref 48: journal name, volume, page numbers are missing.
Author Response
Please check the format of the newly added references!
Response: we have checked the format carefully and corrected the mistakes.
Ref 42: the volume and the page numbers are missing, the first authors given name should be replaced with its initial.
Response: We apologize for the mistake here and it has been corrected.
Ref 43-44-45 and 65: All initials should be after the surnames.
Response: Corrected as suggested.
Ref 48: journal name, volume, page numbers are missing.
Response: Corrected as suggested.